# Activation of CD8 T cells accelerates anti-PD-1 antibody-induced psoriasis-like dermatitis through IL-6

Ryota Tanaka [1], Yuki Ichimura[1], Noriko Kubota[1], Akimasa Saito[1], Yoshiyuki Nakamura[1], Yosuke Ishitsuka[1], Rei Watanabe[2], Yasuhiro Fujisawa[1], Mirei Kanzaki[3], Seiya Mizuno[4], Satoru Takahashi [4], Manabu Fujimoto[2] & Naoko Okiyama [1]✉

Use of immune checkpoint inhibitors that target programmed cell death-1 (PD-1) can lead to various autoimmune-related adverse events (irAEs) including psoriasis-like dermatitis. Our observations on human samples indicated enhanced epidermal infiltration of CD8 T cells, and the pathogenesis of which appears to be dependent on IL-6 in the PD-1 signal blockade-induced psoriasis-like dermatitis. By using a murine model of imiquimod-induced psoriasis-like dermatitis, we further demonstrated that PD-1 deficiency accelerates skin inflammation with activated cytotoxic CD8 T cells into the epidermis, which engage in pathogenic cross-talk with keratinocytes resulting in production of IL-6. Moreover, genetically modified mice lacking PD-1 expression only on CD8 T cells developed accelerated dermatitis, moreover, blockade of IL-6 signaling by anti-IL-6 receptor antibody could ameliorate the dermatitis. Collectively, PD-1 signal blockade-induced psoriasis-like dermatitis is mediated by PD-1 signaling on CD8 T cells, and furthermore, IL-6 is likely to be a therapeutic target for the dermatitis.

[1] Department of Dermatology, Faculty of Medicine, University of Tsukuba, Ibaraki, Japan. [2] Department of Dermatology, Graduate School of Medicine Faculty of Medicine, Osaka University, Osaka, Japan. [3] Department of Dermatology, Mito Saiseikai General Hospital, Ibaraki, Japan. [4] Laboratory Animal Resource Center, Faculty of Medicine, University of Tsukuba, Ibaraki, Japan. ✉email: naoko.okiyama@md.tsukuba.ac.jp

For cancer immune therapies that regulate T cells to enhance immune responses, T cells must successfully recognize tumor antigens through their T-cell receptors (TCRs) and become activated in order to expel tumors[1,2]. In addition, a number of stimulatory and inhibitory receptor and ligand pairs expressed on T cells, antigen-presenting cells (APCs) or tumor cells, termed immune checkpoints, also play crucial roles for both T cell activation and inhibition[3]. Programmed cell death-1 (PD-1) is one of these immune checkpoint molecules, which was initially detected in activated murine T cells upon TCR engagement[4] and subsequently in exhausted T cells[5]. Its ligands, programmed cell death-ligand 1 (PD-L1) and PD-L2, are expressed on various cell types, including hematopoietic cells infiltrating tumors, including APCs, and on non-hematopoietic cells such as cancer cells[6,7]. The interaction between PD-1 and its ligands reduces T cell function by inducing exhaustion, apoptosis, anergy, and downregulation of cytokine production by T cells, leading to suppression of the antitumor immune response[8,9].

In melanoma, PD-1 expression is detected on tumor-infiltrating lymphocytes including tumor antigen–specific T cells, which are functionally impaired. Moreover, the biological activity of these cells can be partially recovered by inhibiting the PD-1 pathway[10–12]. Indeed, anti-PD-1 blocking antibodies such as nivolumab and pembrolizumab function as immune checkpoint inhibitors, and have proven effective for the treatment of melanoma[13,14]. However, as the PD-1 pathway also maintains peripheral T cell tolerance and regulates inflammation[15], inhibition of this pathway may lead to autoimmune manifestations referred to as immune-related adverse events (irAEs)[16,17]. Early clinical trials and reviews have reported that anti-PD-1 antibody-related irAEs occur in more than 70% of patients, and cutaneous irAEs are the most frequently observed (approximately 40%). Further, most cutaneous irAEs are mild (low-grade) and manageable with topical steroids[16,18–21]. On the other hand, it has also been recently reported that two-thirds of patients with cutaneous irAEs reportedly required systemic corticosteroids for the treatment of eruptions, and 19% of patients discontinued cancer-immunotherapy due to irAEs, even though 75% experienced antitumor responses with the therapy[22]. High-dose and/or long-term use of systemic immunosuppressive therapies are required to control such irAEs[23], potentially resulting in prolonged interruption of cancer treatment. Moreover, these immunosuppressive therapies may also abrogate the antitumor response by counteracting lymphocyte activation[20,24]. Therefore, more efficacious, systemic therapies that resolve the symptoms of irAEs while also enabling shorter interruptions of cancer treatments and do not interfere with their antitumor effects would be ideal. In addition, a recent American Society of Clinical Oncology guideline suggests that cutaneous irAEs are increasingly recognized as a contributing factor to treatment noncompliance, discontinuation, or dose modification[24]. Plausibly, such skin manifestations cause changes in appearance along with discomfort, which reduces patient quality of life and results in loss of treatment motivation. We previously reported a case of nivolumab-induced psoriasis-like dermatitis[25], which has been reported to develop in patients treated with anti-PD-1/PD-L1 antibody[25,26]. The latest post-marketing surveillance of nivolumab in Japan reports that 2,391 cases of cutaneous irAE occurred, of which 103 cases (4.3 %) were labeled as psoriasis. Notably, more than 18% (19 /103) of those cases were reportedly severe[27]. Importantly, the mechanism by which psoriasis-like dermatitis occurs following PD-1/PD-L1 inhibition remains unknown, and strategies to mitigate the occurrence of especially severe cases are yet to be identified. With the recent increase in use of anti-PD-1 antibody for patients with various types of cancers, clarification of the underlying mechanisms and development of more efficacious treatment for PD-1 signal blockade-induced psoriasis-like dermatitis is needed.

Application of imiquimod (IMQ), a toll-like receptor 7/8 agonist, is known to induce psoriasis-like dermatitis in both humans[28] and mice[29]. Furthermore, it has already been reported that both PD-1 genetic deficiency and blockade of PD-1 with a specific monoclonal antibody exacerbate IMQ-induced psoriasis-like dermatitis in mice[30]. Therefore, it is likely that the pathophysiological mechanism of PD-1 signal blockade-induced psoriasis-like dermatitis could be elucidated using this murine model.

The present study aimed to elucidate the characteristics and mechanisms underlying psoriasis-like dermatitis induced by blocking PD-1 signaling, and to identify suitable treatments. The observations from human samples and further experiments using a preclinical murine model of IMQ-induced psoriasis-like dermatitis demonstrated that the dermatitis was accelerated by an increase of skin-infiltrating activated, cytotoxic CD8 T cells allowing pathogenic crosstalk with keratinocytes and subsequent production of IL-6. Moreover, blockade of interleukin (IL)-6 signaling by anti-IL-6 receptor blocking antibody (MR16-1) restrained the PD-1 signal blockade provoked by severe dermatitis by inhibiting both Th17 cell differentiation and cytotoxic CD8 T cell activation. Thus, this highlights the significance of IL-6 blockade therapy specifically for the regulation of PD-1 signal blockade-induced dermatitis.

## Results

**Increased CD8/CD4 ratio of epidermal-infiltrating lymphocytes in cases of anti-PD-1 antibody-induced psoriasis-like dermatitis compared to cases of idiopathic psoriasis.** Immunohistochemical (IHC) evaluation of skin biopsy samples, as demonstrated in Fig. 1a, revealed that CD8/CD4 ratios of epidermal-infiltrating mononuclear cells were significantly increased in cases of anti-PD-1 antibody-induced psoriasis-like dermatitis (median ± standard deviation [SD], $3.48 ± 1.0$) compared to that in cases of idiopathic psoriasis ($1.06 ± 0.19$, $P = 0.008$ by Mann–Whiney $U$ test, Fig. 1b).

**Elevated serum IL-6 correlates with the development of anti-PD-1 antibody-induced psoriasis-like dermatitis in humans.** We reported in our preliminary study that only increased serum levels of IL-6, but not those of IL-17A, interferon (IFN)-γ and IL-8, correlated with the development of anti-PD-1 antibody-induced psoriasis-like dermatitis in patients with malignant melanoma[25]. In order to validate this phenomenon, we analyzed the serum levels of IL-6 in eight cases of psoriasis-like dermatitis, and 19 cases without any irAEs. Cases of psoriasis-like dermatitis exhibited significantly higher serum IL-6 levels compared to those of IL-6 in cases without any irAEs ($P < 0.0001$ by Mann–Whiney $U$ test, Fig. 1c). Our additional analysis using the remaining samples showed that there was no significant difference in the serum levels of soluble IL-6 receptor alfa (sIL-6Rα) between the two groups, six cases of psoriasis-like dermatitis and 18 cases without any irAEs (Supplemental Fig. 1). Collectively, these results suggest that the pathogenesis of anti-PD-1 antibody-induced psoriasis-like dermatitis may depend on IL-6.

**PD-1$^{-/-}$ mice exhibit more severe IMQ-induced psoriasis-like dermatitis than WT mice.** PD-1$^{-/-}$ mice developed significantly more severe IMQ-induced psoriasis-like dermatitis (Fig. 2a) when compared to WT mice as revealed by clinical measurements including ear swelling (change from the baseline at day 7, $20.6 ± 2.6$ μm vs. $7.2 ± 1.4$ μm, $P = 0.0014$ by two-way ANOVA, Fig. 2b) and PASI score, which represents the severity of erythema, scaling

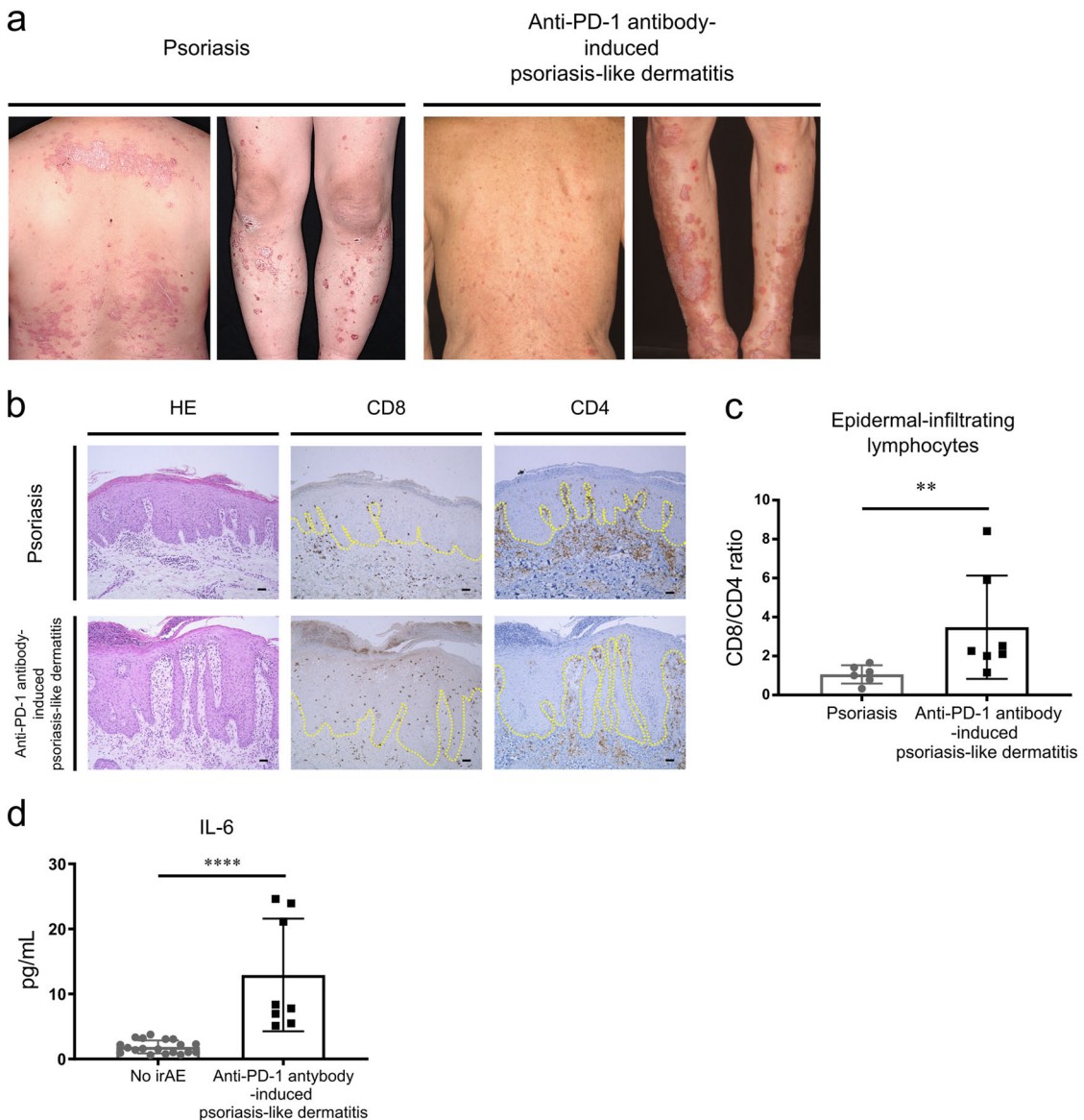

**Fig. 1 Characteristics of anti-programmed cell death-1 (PD-1) antibody-induced psoriasis-like dermatitis. a** Representative clinical images of patients with idiopathic psoriasis and anti-PD-1 antibody-induced psoriasis-like dermatitis. Both patients developed well-defined scaly plaques scattered over their trunks and extremities. **b** Representative hematoxylin and eosin (HE)-stained, and anti-CD8 or CD4 antibody-stained skin biopsy samples from patients with idiopathic psoriasis and anti-PD-1 antibody-induced psoriasis-like dermatitis. Scale bars = 50 μm. **c** CD8/CD4 ratios of epidermal-infiltrating lymphocytes ($n = 6$ and 7 in idiopathic psoriasis and anti-PD-1 antibody-induced psoriasis-like dermatitis, respectively). $**P < 0.01$ by nonparametric 2-tailed Mann–Whitney $U$ test. d Profiles of serum interleukin (IL)-6 levels in serum samples from anti-PD-1 antibody-treated cancer patients who developed psoriasis-like dermatitis as an immune-related adverse event (irAE, $n = 8$) and those with no irAE ($n = 19$). $****P < 0.0001$ by nonparametric 2-tailed Mann–Whitney $U$ test.

and skin thickness, ($7.8 \pm 0.2$ vs. $3.2 \pm 0.2$, $P < 0.0001$ by two-way ANOVA, Fig. 2c) at day 7. Moreover, pathological analysis, as shown in Fig. 2d, of epidermal thickness ($61.5 \pm 7.9$ μm vs. $35.6 \pm 3.2$ μm, $P = 0.008$ by Mann–Whitney $U$ test, Fig. 2e) and the number of epidermal neutrophilic micro-abscesses ($3.2 \pm 0.58$/ear slide vs. WT $0.6 \pm 0.24$/ear slide, $P = 0.008$ by Mann–Whitney $U$ test, Fig. 2f) at day 7 further indicates the protective role of PD-1 in IMQ-induced psoriasis-like dermatitis.

Moreover, we confirmed that mice treated with anti-PD-1 blocking monoclonal antibody developed clinically and histo-pathologically severe IMQ-induced psoriasis-like dermatitis compared to control mice treated with isotype IgG2a control

(Ctrl) antibody (Supplemental Fig. 2A–E). The results corresponded to the experiments using PD-1$^{-/-}$ mice.

We also conducted an experiment using the B16 melanoma murine model, in which B16F10 melanoma cells were inoculated into backs of both WT and PD-1$^{-/-}$ mice, to investigate whether the presence of cancer involves PD-1 blockade-induced psoriasis-like dermatitis. First, this model did not induce psoriasis-like dermatitis spontaneously nor under vehicle cream treatment (Supplemental Fig. 3A–E). Moreover, the presence of B16 melanoma did not lead to a significant difference in PASI score or ear swelling in WT or PD-1$^{-/-}$ mice (Supplemental Fig. 3A–H).

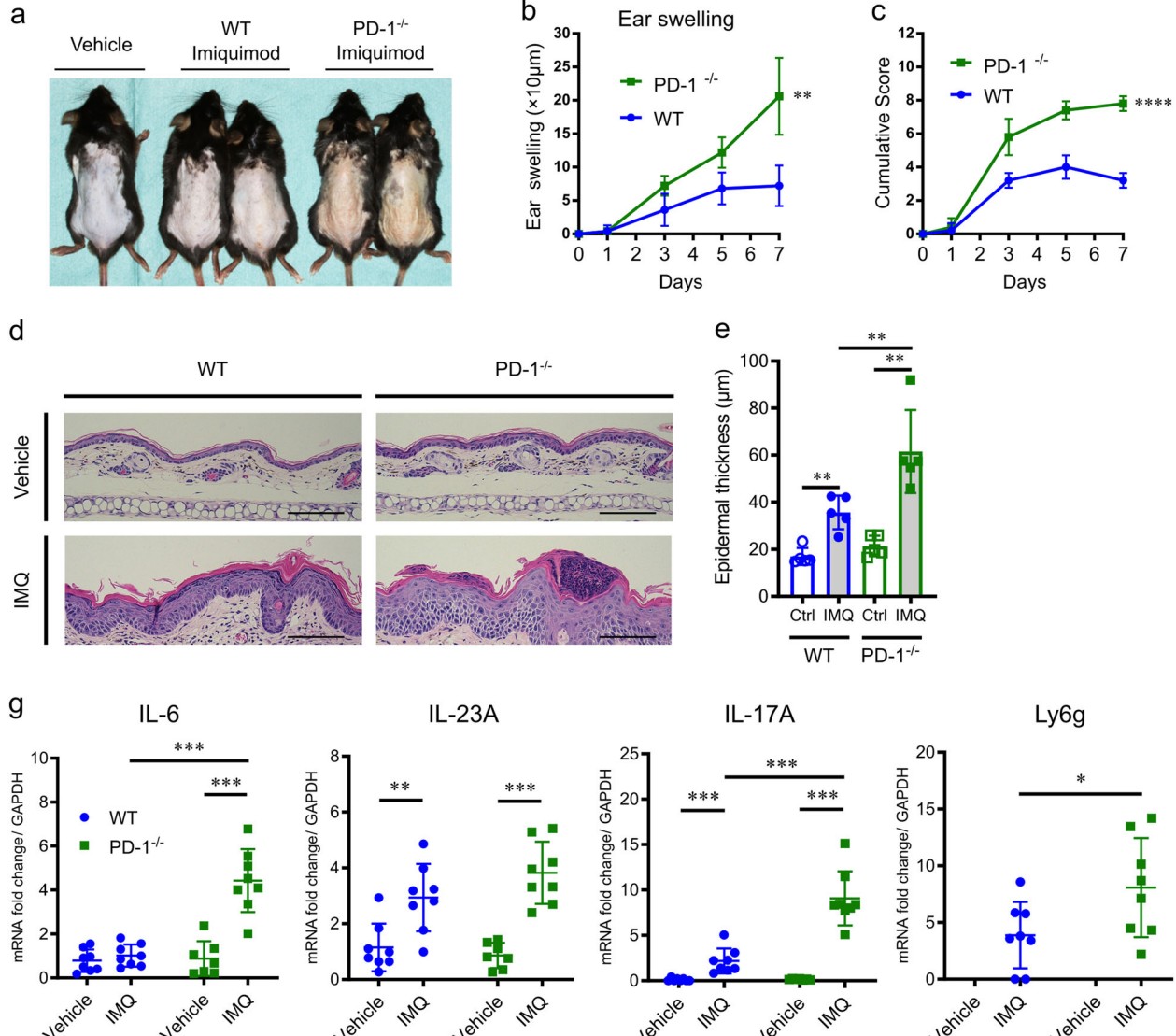

**Fig. 2 Comparison of clinical and histological appearance and cytokine mRNA expression in imiquimod (IMQ)-induced psoriasis-like dermatitis between PD-1$^{-/-}$ mice and wild-type (WT) mice. a** Representative clinical images at day 7 of IMQ-induced psoriasis-like dermatitis in WT and PD-1$^{-/-}$ mice. Application of vehicle cream was used as a control. **b**, **c** The course of ear swelling (**b**) and PASI score (**c**) representing the severity of erythema, scaling, and skin thickness of WT and PD-1$^{-/-}$ mice. ** $P<0.01$ and **** $P<0.0001$ by two-way ANOVA. **d** Representative images of HE-stained ear samples from IMQ-induced psoriasis-like dermatitis in WT and PD-1$^{-/-}$ mice at day 7. Application of vehicle cream was used as a control. Scale bars, 100 μm. **e**, **f** Epidermal hyperplasia (**e**), and the number of epidermal neutrophilic micro-abscesses (**f**) in the ear samples from IMQ-applied WT or PD-1$^{-/-}$ mice ($n=5$ in each group). Data are shown as mean ± standard deviation (SD). Data are representative of three independent experiments. **$P<0.01$ by nonparametric 2-tailed Mann–Whitney $U$ test. **g** Quantitative reverse transcriptase-polymerase chain reaction (qRT-PCR) analysis of psoriasis-related cytokines and the neutrophilic surface marker Ly6g in ear samples from IMQ- or vehicle-treated WT and PD-1$^{-/-}$ mice at day 7 ($n=7$–8 in each group). Fold changes in mRNA levels were calculated and normalized against GAPDH mRNA levels. Data are expressed as mean ± SD. Data are representative of two independent experiments. *$P<0.05$, **$P<0.01$, and ***$P<0.001$ by nonparametric 2-tailed Mann–Whitney $U$ test.

Taken together, these data suggest that PD-1 blockade, either by genetic knockout or antibody treatment, promotes IMQ-induced psoriasis-like dermatitis, and that PD-1 blockade in the context of cancer does not increase the severity of dermatitis.

**PD-1 deficiency in mice results in increased epidermal infiltration of CD8 T cells with enhanced production of IFN-γ and CXC chemokine ligand (CXCL9).** IHC analysis of murine ear skin samples revealed significantly increased numbers of CD8 T cells infiltrating into the epidermis of PD-1$^{-/-}$ mice and anti-PD-1 antibody-treated mice when compared to control mice (Fig. 3a and b, $P=0.008$ by Mann–Whitney $U$ test, and

Supplemental Fig. 2F, $P=0.008$ by Mann–Whitney $U$ test), similar to what was seen in the patients with anti-PD-1 antibody-induced psoriasis-like dermatitis. Next, qRT-PCR analysis revealed that PD-1$^{-/-}$ mice have significantly higher CD8a and IFN-γ mRNA levels in CD45$^+$ epidermal cells and CXCL9 in keratinocytes (CD45-negative epidermal cells) compared to that of WT mice (Fig. 3c, $P=0.008$, $P=0.008$ and $P=0.03$ by Mann–Whitney $U$ test, respectively).

**PD-1 on CD8 T cells regulates the development of IMQ-induced psoriasis-like dermatitis.** Following 7 days of daily IMQ application, PD-1-cKO (PD-1$^{fl/fl}$CD8$^{Cre}$) mice were found to

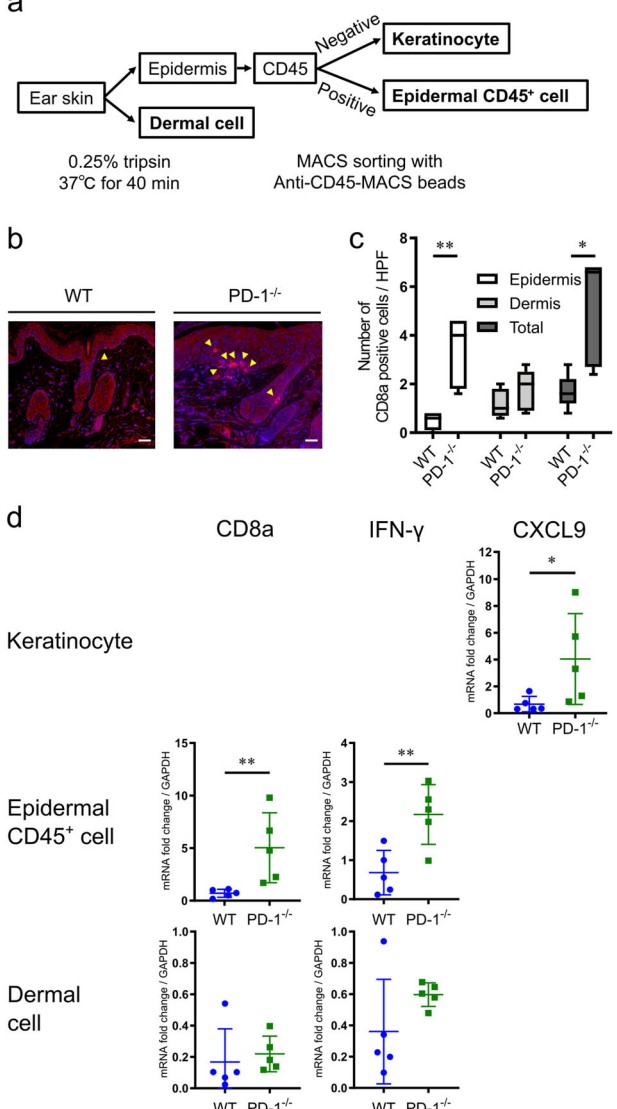

**Fig. 3 PD-1 deficient CD8 T cells display enhanced interferon (IFN)-γ production, and IFN-γ-stimulated keratinocytes produce CXC chemokine ligand (CXCL)9 in IMQ-induced psoriasis-like dermatitis. a** A schematic illustrating the protocol for processing of ear skin samples into keratinocyte, epidermal CD45+ cell and dermal cell populations. **b** Representative images of immunohistochemical (IHC) staining of CD8 T cells in IMQ-applied WT and PD-1−/− mice at day 5. Scale bars, 20 μm. **c** The number of infiltrated CD8 T cells in the epidermis. **d** The qRT-PCR analysis of CD8a, IFN-γ, and CXCL9. Data are from five mice per group, and are representative of two independent experiments. *P < 0.05 and **P < 0.01 by nonparametric 2-tailed Mann–Whitney U test.

have developed more severe IMQ-induced psoriasis-like dermatitis than littermate Ctrl (PD-1fl/+CD8Cre) mice (Fig. 4c), when evaluated clinically by the change in ear swelling from the baseline to day 7 (17 ± 1.4 μm vs. 6.8 ± 0.75 μm, P < 0.0001 by two-way ANOVA, Fig. 4d), PASI score at day 7 (6.3 ± 0.42 vs. 3.8 ± 0.25, P = 0.0001 by two-way ANOVA, Fig. 4e), and pathologically by epidermal thickness at day 7 (59.5 ± 2.6 μm vs. 37.4 ± 3.7 μm, P = 0.008 by Mann–Whitney U test, Fig. 4f, g). qRT-PCR analysis revealed that PD-1-cKO mice showed significantly higher levels of both CD8a and IFN-γ mRNA in serum than that found in littermate Ctrl mice (P = 0.015 and P = 0.015 by Mann–Whitney U test, respectively, Fig. 4h). The number of CD8 T cells (CD45

+CD3+CD8a+ cells) in draining lymph nodes (dLNs) was increased, and more CD8 T cells in PD-1-cKO mice produced IFN-γ and Gzm B than that of littermate Ctrl mice (Fig. 4i, j).

In summary, these in vivo results suggest that the PD-1 deficiency enhances the numbers of infiltrating activated cytotoxic CD8 T cells, resulting in acceleration of psoriasis-like dermatitis.

**Enhanced expression of cutaneous IL-6 via PD-1 deficiency in mice.** qRT-PCR analysis revealed that unstimulated ear skin from PD-1−/− and WT mice contain similar low levels of psoriasis-related cytokines, IL-6, IL23-A, and IL-17A (P = 0.95, P = 0.57 and P = 0.21 by Mann–Whitney U test, respectively, Fig. 2g). IMQ application significantly increased mRNA expression of IL-23A and IL-17A in both WT mice and PD-1−/− mice (P = 0.007 and P = 0.0002 in WT mice, and P = 0.0003 and P = 0.0003 in PD-1−/− mice by Mann–Whitney U test, respectively, Fig. 2g). Notably, increased IL-6 mRNA expression induced by IMQ application was observed only in PD-1−/− mice and not in WT mice (P = 0.0006 and P = 0.27, respectively, by Mann–Whitney U test, Fig. 2g). Expression of Ly6g, a neutrophil surface marker, mRNA was undetectable in both groups after vehicle cream application, but were increased significantly in PD-1−/− mice compared to WT mice after IMQ application (P = 0.048 by Mann–Whitney U test, Fig. 2g). In addition, these results were also confirmed using PD-1-specific blocking antibody treatment (Supplemental Fig. 2G). Further investigations revealed a significantly higher level of IL-6 mRNA expression in the CD45-positive epidermal cells, and an increased total number of CD45-positive epidermal cells with specific infiltration of neutrophils in PD-1−/− mice compared to WT mice (Supplemental Fig. 4).

Collectively, IL-6 expression related to expression of Th17 cytokines and infiltration of neutrophils correlates with PD-1 deficiency-enhanced IMQ-induced psoriasis-like dermatitis.

**Blockade of the IL-6R ameliorates PD-1 deficiency-exacerbated psoriasis-like dermatitis.** The increase of serum IL-6 levels post-treatment with anti-PD-1 blocking antibody implies that the pathogenesis of PD-1 signal blockade-induced psoriasis-like dermatitis is dependent on IL-6. Therefore, we employed blockade of IL-6 signaling using an anti-IL-6R blocking antibody (MR16-1) in order to assess the effects on IMQ-induced psoriasis-like dermatitis. The baseline serum levels of psoriasis-related cytokines (IL-6, IL-17A, and IL-23A) were the same between WT and PD-1−/− mice (n = 3). Induction of psoriasis-like dermatitis by IMQ elevated these cytokines in both IgG Ctrl-treated WT and PD-1−/− mice, and markedly in mice with PD-1 deficiency (Fig. 5h). MR16-1-treated PD-1−/− mice showed significantly less IMQ-induced psoriasis-like dermatitis compared to IgG Ctrl-treated PD-1−/− mice as evaluated by ear swelling (7.0 ± 0.6 μm vs. 15.7 ± 1.8 μm, P = 0.0013 by two-way ANOVA) and PASI score (4.5 ± 0.6 vs. 7.3 ± 0.8, P = 0.005 by two-way ANOVA) at day 7. Moreover, MR16-1-treated PD-1−/− mice were clinically similar to IgG Ctrl-treated WT mice (ear swelling 7.8 ± 0.6 μm and PASI 4.3 ± 0.5; P = 0.56 and P = 0.10 by two-way ANOVA, respectively, Fig. 5a–c). Histological analyses also revealed that MR16-1-treated PD-1−/− mice had less severe psoriasis than IgG Ctrl-treated PD-1−/− mice with reduced epidermal hyperplasia (38.8 ± 2.0 μm vs. 55.8 ± 1.8 μm, P = 0.01 by Mann–Whitney U test, Fig. 5d, e) and reduced numbers of epidermal neutrophilic micro-abscesses (3.2 ± 0.4 vs. 6.2 ± 0.6, P = 0.008 by Mann–Whitney U test), which was the same as IgG Ctrl-treated WT mice (34.8 ± 2.5 μm, P = 0.06; and 2.4 ± 0.4, P = 0.32 by Mann–Whitney U test, respectively, Fig. 5f). Further, compared to IgG Ctrl-treated PD-1−/− mice, MR16-1-treated PD-1−/−

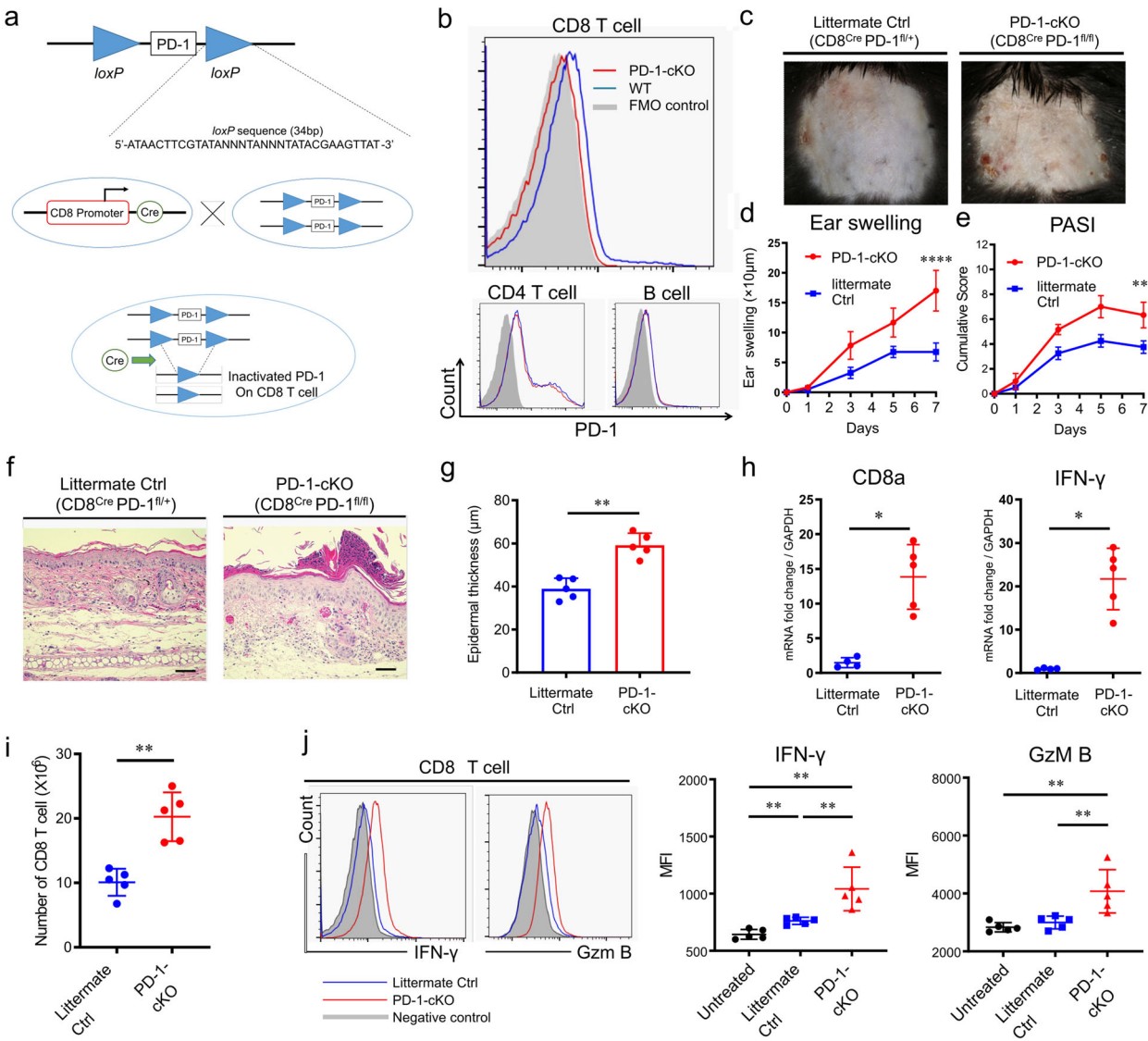

**Fig. 4 Clinical and histological evaluation of IMQ-induced psoriasis-like dermatitis in conditional knockout mice with PD-1 deficiency specifically in CD8 T cells. a** An overview of the PD-1-floxed mouse and the breeding strategy for conditional mutation using loxP and cyclization recombinase (Cre) driving mouse lines. Specific 34bp DNA fragments representing the loxP (locus of x-over, P1) sites were inserted across the *PD-1* gene (Top). Conditional knockout (cKO) mice were generated by breeding the CD8a-Cre knock-in mouse strain with the PD-1-floxed mouse strain (Bottom). **b** Specific deletion of PD-1 expression in CD8 T cell, but not in CD4 T cells or B cells, of IMQ-treated PD-1-cKO mice (red line) compared to IMQ-treated WT littermates (blue line). Fluorescence Minus One (FMO) was used as a control (gray area). Data represent three independent experiments. **c** Representative clinical images of psoriasis-like dermatitis in PD-1cKO (CD8$^{Cre}$ PD-1$^{fl/fl}$) mice and littermate control (Ctrl, CD8$^{Cre}$ PD-1$^{fl/+}$) mice at day 7. **d, e** Ear swelling (**d**) and PASI score ($n = 4$ in each group). Data are representative of three independent experiments. ***$P<0.001$ and ****$P<0.0001$ by two-way ANOVA. **f** Representative images of HE-stained ear skin samples from IMQ-treated littermate Ctrl mice and PD-1-cKO mice at day 7. Scale bars, 100 μm. **g** Epidermal thickness ($n = 5$ in each group). Data are representative of three independent experiments. **h** qRT-PCR analysis of CD8a and IFN-γ mRNA levels in ear skin samples from IMQ-treated littermate Ctrl mice ($n = 4$) and PD-1-cKO mice ($n = 5$) at day 7. Fold changes in mRNAs levels were normalized against GAPDH mRNA levels. **i** Total numbers of CD8 T cells in draining lymph nodes (dLNs) from IMQ-treated littermate Ctrl mice and PD-1-cKO mice at day 7 ($n = 5$ in each group). **j** Representative histograms of IFN-γ and Granzyme B (GzmB) production by CD8 T cells in the dLNs. The gray histograms represent negative controls. Graphs of median fluorescent intensities (MFIs) of IFN-γ and Gzm B. The results are presented as means ± SDs. Data are representative of two independent experiments. *$P<0.05$ and **$P<0.01$ by nonparametric 2-tailed Mann–Whitney $U$ test.

mice presented significantly suppressed levels of psoriasis-related cytokine mRNAs IL-6, IL-17a, and IL-23a in the ear skin at day 7 ($P = 0.003$, $P = 0.03$ and $P = 0.02$, respectively, by Mann–Whitney $U$ test, Fig. 5g), and significantly decreased serum levels of IL-17A and IL-23A ($P = 0.03$ by Mann–Whitney $U$ test) to the baseline level at day 7 (Fig. 5h). These cytokine expression levels (IL-6, IL-17A, and IL-23A) in MR16-1-treated PD-1$^{-/-}$

mice were the same as those seen in IgG Ctrl-treated WT mice ($P = 0.44$, $P = 0.21$ and $P = 0.66$ in skin mRNA levels, respectively, and $P = 0.57$, $P = 0.15$ and $P = 0.15$ in serum levels, respectively, analyzed by Mann–Whitney $U$ test).

Furthermore, we employed blockade of IL-17A signaling using an anti-IL-17A neutralizing monoclonal antibody in order to compare the effect with anti-IL-6R antibody on IMQ-induced

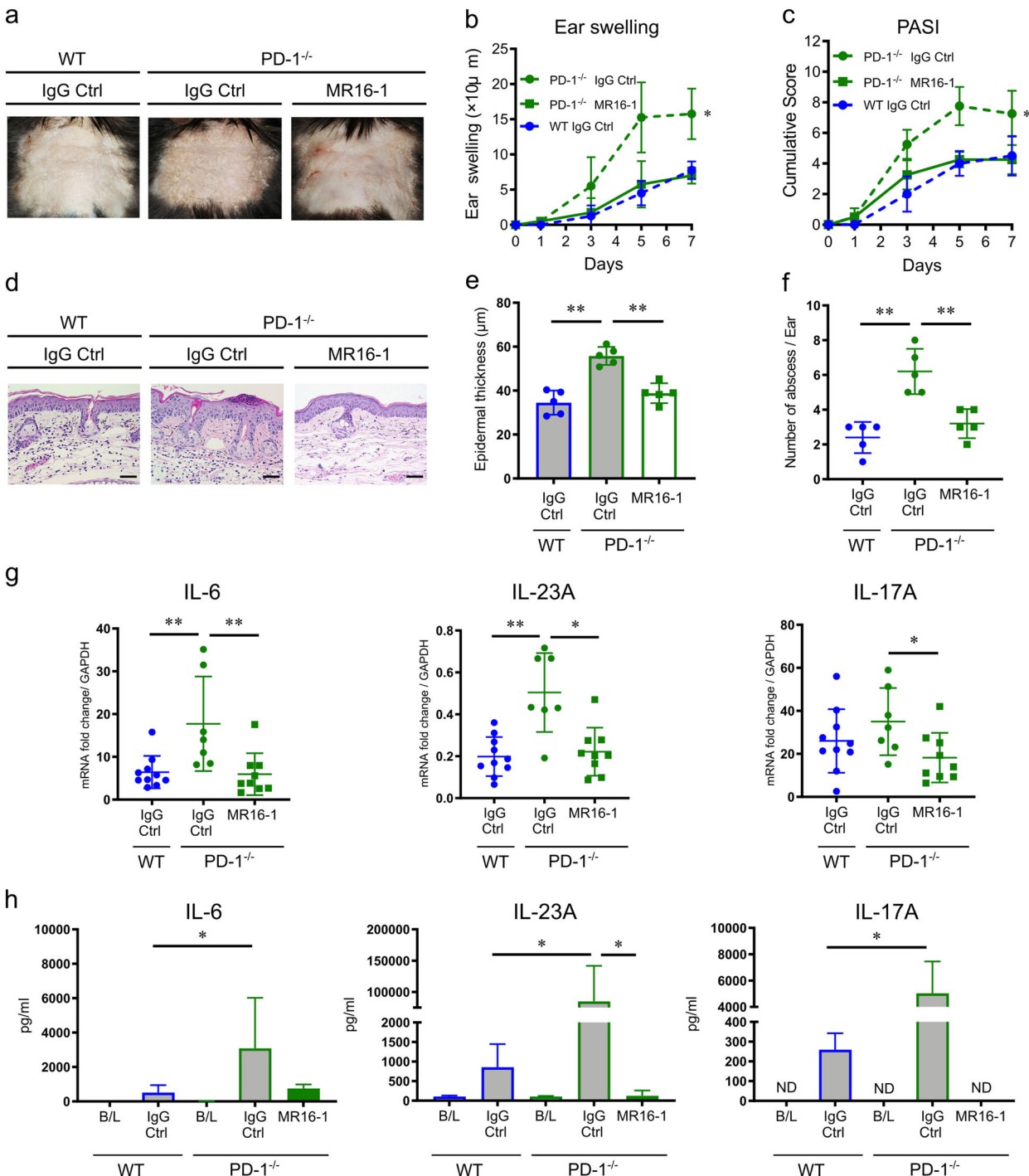

**Fig. 5 Characteristics of anti-IL-6 receptor (IL-6R) antibody-treated IMQ-induced psoriasis-like dermatitis in PD-1$^{-/-}$ mice. a** Representative clinical images of anti-IL-6R antibody (MR16-1)- or IgG Ctrl-treated IMQ-induced psoriasis-like dermatitis in PD-1$^{-/-}$ mice compared to IgG Ctrl-treated WT mice at day 7. **b** Ear swelling. **c** PASI score ($n$=4 in each group). Data are representative of two independent experiments. * $P$<0.05 and ** $P$<0.01 by two-way ANOVA. **d** Representative HE staining of ear skin samples from IgG Ctrl-treated WT mice, IgG Ctrl- or MR16-1-treated PD-1$^{-/-}$ mice at day 7 ($n$=5 in each group). Scale bars, 50 μm. **e** Epidermal thickness ($n$=5 in each group). **f** The number of epidermal, neutrophilic micro-abscess ($n$=5 in each group). **g** qRT-PCR analysis of mRNA expression levels of psoriasis-related cytokines, IL-6, IL-23a, and IL-17a, in ear skin samples from IgG Ctrl-treated WT mice ($n$=10), IgG Ctrl-treated PD-1$^{-/-}$ mice ($n$=7), and MR16-1-treated PD-1$^{-/-}$ mice ($n$=9) at day 7 for IMQ application. Fold changes in mRNA levels normalized to GAPDH mRNA levels. **h** Multiplex, bead-based analysis of serum levels of psoriasis-related cytokines, IL-6, IL-23A, and IL-17A in IgG Ctrl-treated WT mice ($n$=4), IgG Ctrl-treated PD-1$^{-/-}$ mice ($n$=4) and MR16-1-treated PD-1$^{-/-}$ mice ($n$=5) at day 7 for IMQ application. Baseline (B/L) serum levels of these cytokines were also measured ($n$=3 each). In some samples, cytokines were not detected (ND). Data are expressed as mean ± SEM. Data are representative of two independent experiments. *$P$<0.05 and **$P$<0.01 by nonparametric 2-tailed Mann–Whitney $U$ test.

psoriasis-like dermatitis in PD-1$^{-/-}$ mice. When evaluated both clinically and histologically at day 7, treatment with anti-IL-17A antibody improved the dermatitis in both PD-1$^{-/-}$ and WT mice to a level equivalent to treatment with anti-IL-6R antibody (Supplemental Fig. 5A–E). However, anti-IL-17A antibody-treated PD-1$^{-/-}$ mice showed more severe IMQ-induced psoriasis-like dermatitis than IgG Ctrl-treated WT mice, as evaluated by PASI score (4.8 ± 0.8 vs. 3.6 ± 0.5, $P = 0.003$ by two-way ANOVA) and ear swelling (7.4 ± 1.6 µm vs. 5.2 ± 1.4 µm, $P = 0.053$ by two-way ANOVA) at day 5 (Supplemental Fig. 5B). In contrast, anti-IL-6R antibody treatment improved the dermatitis in PD-1$^{-/-}$ mice earlier, at day 5 (Fig. 5b, c). These results indicate the delayed efficacy of anti-IL-17A neutralizing antibody treatment, compared to treatment with anti-IL-6R antibody, for PD-1 signal blockade-induced psoriasis-like dermatitis.

Taken together, blockade of IL-6 signaling with an anti-IL-6R antibody is a potential therapeutic approach to resolve psoriasis-like dermatitis caused by inhibition of PD-1. Moreover, the treatment kinetics of anti-IL-6R antibody appear to be shorter than that of anti-IL-17A antibody treatment.

**Accelerated psoriasis-like dermatitis due to PD-1 deficiency on CD8 T cells can be ameliorated by the treatment with anti-IL-6R blocking antibody.** To investigate whether blockade of IL-6 with anti-IL-6R antibody could restrain PD-1 deficiency-induced activation of CD8 T cells, both PD-1-cKO mice and their littermate Ctrl mice were treated with either anti-IL-6R antibody (MR16-1) or isotype IgG control. MR16-1-treated PD-1-cKO mice exhibited significant improvement in clinical manifestations of IMQ-induced psoriasis-like dermatitis compared to IgG Ctrl-treated PD-1-cKO mice (ear swelling change from the baseline on day 7, 7.8 ± 1.5 µm vs. 16.3 ± 1.4 µm, $P = 0.0012$ by two-way ANOVA; and PASI score at day 7, 4.3 ± 0.2 vs. 8.2 ± 0.7, $P < 0.0001$ by two-way ANOVA). Moreover, this was to a similar level as that of IgG Ctrl-treated littermate Ctrl mice (ear swelling 6.8 ± 1.4 µm, $P = 0.85$ by two-way ANOVA; and PASI 3.6 ± 0.4, $P = 0.66$ by two-way ANOVA, Fig. 6a–c). Histological evaluation also indicated that MR16-1-treated PD-1-cKO mice had less epidermal hyperplasia and reduced numbers of epidermis-infiltrating CD8 T cells than did IgG Ctrl-treated PD-1-cKO mice (48.7 ± 3.1 µm vs. 74.0 ± 3.7 µm, $P = 0.002$ by Mann–Whitney $U$ test, Fig. 6d, E; 20.8 ± 6.9 vs. 73.8 ± 15.2, $P = 0.015$ by Mann–Whitney $U$ test). In fact, the response in MR16-1-treated PD-1-cKO mice occurred at the same level as that of IgG Ctrl-treated littermate Ctrl mice (43.7 ± 1.0 µm, $P = 0.25$ by Mann–Whitney $U$ test; and 8.4 ± 2.8, $P = 0.16$ by Mann–Whitney $U$ test, Fig. 6d, e). Furthermore, MR16-1-treated PD-1-cKO mice displayed significantly suppressed numbers of CD8 T cells in the dLNs and reduced *CD8a* and IFN-γ mRNA levels in the ear skin samples at day 7 when compared to IgG Ctrl-treated PD-1-cKO mice ($P = 0.004$, $P = 0.03$ and $P = 0.09$, respectively, by Mann–Whitney $U$ test, Fig. 5g, h).

Importantly, there were not any differences between MR16-1-treatment and IgG Ctrl-treatment in littermate Ctrl mice, highlighting the significance of IL-6 blockade therapy for the regulation of PD-1 signal blockade-activated CD8 T cells in psoriasis-like dermatitis.

## Discussion

The pathogenesis of cutaneous irAEs in patients treated with anti-PD-1 antibody has yet to be elucidated. However, previous reports suggest that activated proliferative intradermal CD8 T cells evoke cutaneous irAEs such as lichen planus-like dermatitis and eczematous reaction[31,32]. The present study highlights the importance of PD-1 expression on CD8 T cells for the regulation of psoriasis-

like dermatitis. We found that CD8-positive lymphocyte infiltration into the epidermis was significantly increased in patients with anti-PD-1 antibody-induced psoriasis-like dermatitis compared to that in idiopathic psoriasis. A murine model of IMQ-induced psoriasis-like dermatitis clearly demonstrated that PD-1 deficiency accelerates infiltration of epidermal CD8 T cells with enhanced IFN-γ production of inflamed skin, and IFN-γ-stimulated keratinocytes produced an IFN-γ-inducible chemokine (CXCL9) for recruitment of T cells. Furthermore, the newly generated cKO mice with PD-1 deficiency specifically in CD8-positive cells demonstrated more severe IMQ-induced psoriasis-like dermatitis compared to the littermate control mice. These results suggest that PD-1 regulates skin-infiltrating CD8 T cells to engage in pathogenic crosstalk with PD-L1 expressed on various cells including keratinocytes[33]. In idiopathic psoriasis activation of conventional dendritic cells producing IL-23 lead to expansion and activation of autoreactive CD8 T cells in the dermis, which in turn acquire expression of α1β1-integrin and migrate into the epidermis. The epidermis has been identified as an ideal location for CD8 T cells to engage in pathogenic crosstalk with keratinocytes[34,35]. Furthermore, intra-epidermal CD8 T cells are shown to be highly pathogenic as the accumulation of epidermal T cells parallels the increase in proliferating keratinocytes in vivo[34]. Collectively, PD-1 signal blockade-induced activation of CD8 T cells is essential to induce and accelerate anti-PD-1 antibody-induced psoriasis-like dermatitis.

We also found a significant increase in the serum levels of IL-6 in patients with anti-PD-1 antibody-induced psoriasis-like dermatitis, as we had shown in our preliminary study[25], indicating that IL-6 could play an important role during disease development and thus, may be a suitable treatment target. As expected, a murine model of IMQ-induced psoriasis-like dermatitis enhanced via PD-1 deficiency was significantly improved by anti-IL-6R blocking antibody. These results clearly show the efficacy of IL-6–targeting therapy for PD-1 deficiency abrogated psoriasis-like dermatitis. One essential role of IL-6 is in the promotion of T helper 17 (Th17) cell production[36]. Th17 cells were recently shown to be a main pathological cell population in idiopathic psoriasis, and blockade of IL-17A and IL-23 have been established as treatments[37], although IL-6 was not established as a potential therapeutic target. Our results also demonstrate that increased production of Th17-related cytokines, such as IL-17A and IL-23A, was accelerated in PD-1$^{-/-}$ mice and was significantly suppressed by IL-6 blockade at both the tissue mRNA and serum levels to the same level as control WT mice.

IL-6 signals through the IL-6Rα and β subunit glycoprotein 130 (gp130). However, as for cells that do not express IL-6Rα on their surface, such as CD8 T cells, trans-signaling, a process whereby IL-6 signaling occurs through a complex of IL-6 and a soluble form of the IL-6Rα binding to ubiquitously expressed gp130[38], is believed to occur. Thus, IL-6 trans-signaling likely plays an important role for the development of cytotoxic CD8 T cell function[39]. Therefore, it is likely that increased levels of soluble IL-6 in PD-1$^{-/-}$ mice promotes cytotoxic CD8 T cell function via IL-6 trans-signaling. Furthermore, our analysis of human samples from anti-PD-1 antibody-treated cancer patients revealed that the serum level of sIL-6 presented correlates with that of sIL-6Rα in patients with anti-PD-1 antibody-induced psoriasis-like dermatitis, which would result in enhanced epidermal infiltration of CD8 T cells. Therefore, blocking this trans-signaling process with anti-IL-6R antibody might decrease the inflammation seen during PD-1 signal inhibition-provoked psoriasis-like dermatitis by impairing the promotion of CD8 T cells. Indeed, mice with PD-1-deficiency specifically in CD8 T cells display severe psoriasis-like dermatitis, which can be restrained by blockade of IL-6 signaling.

Collectively, this treatment strategy comprised of selective blockade of IL-6 signal with anti-IL-6R blocking antibody could be

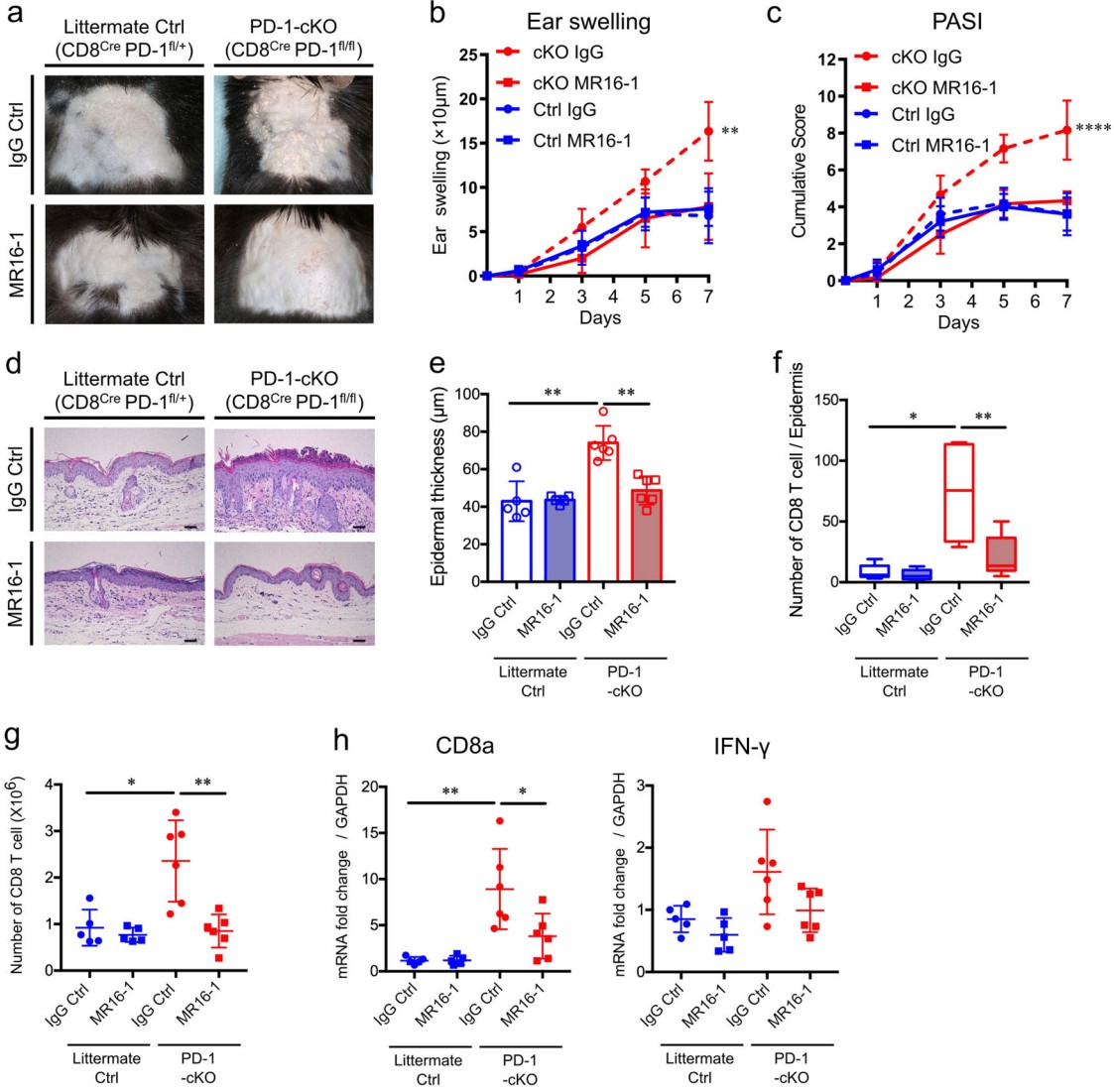

**Fig. 6 Characteristics of anti-IL-6R antibody-treated IMQ-induced psoriasis-like dermatitis in cKO mice with PD-1 deficiency specifically in CD8 T cells.** Representative clinical images of IgG Ctrl- or MR16-1-treated IMQ-induced psoriasis-like dermatitis in littermate Ctrl mice or PD-1-cKO mice. **b** Ear swelling. **c** PASI score. Data are representative of two independent experiments. **$P < 0.01$ and ****$P < 0.0001$ by two-way ANOVA. **d** Representative histological images of HE-stained ear skin samples from these mice at day 7. Scale bars, 50 μm. **e** Epidermal thickness. **f** The number of epidermal, neutrophilic micro-abscess. **g** Total numbers of CD8 T cells in dLNs at day 7. **h** mRNA expression levels of CD8a and IFN-γ in ear skin samples at day 7. Fold changes in mRNAs levels were normalized to GAPDH mRNA levels. $n$ = 5–6 in each group. Data are expressed as mean ± SEM. Data are representative of two independent experiments. *$P < 0.05$, **$P < 0.01$ by nonparametric 2-tailed Mann–Whitney $U$ test.

effective and ideal for treating PD-1 signal blockade-induced psoriasis-like dermatitis. In fact, there have been a few case reports and a retrospective cohort study showing successful treatment of steroid-refractory irAEs with one dose of an anti-human IL-6R antibody, tocilizumab[40–42], even though its use in irAE has not yet been validated. Thus, the present study for the first time demonstrates the rationale for this treatment and the pathophysiology of IL-6 signaling in PD-1 signal inhibition-provoked autoimmunity. On the other hand, a synergistic antitumor effect has been demonstrated on combined blockade of both IL-6 signaling and PD-1/PD-L1 pathways in tumor-bearing mice[43], suggesting the efficacy of the dual signal blockade in terms of resolving the symptoms of irAEs without interfering antitumor effects. Even though psoriasis-like eruptions have been reported as a paradoxical phenomenon after use of tocilizumab[44], our experiments and the previous clinical case report[42] demonstrate that IL-6 blockade

therapy during the initial phase of PD-1 signal blockade-induced psoriasis-like dermatitis may rapidly reduce the severity of irAE and therefore, result in shorter interruptions of cancer treatments. Collectively, individuals with PD-1 signal blockade-induced psoriasis-like dermatitis can potentially benefit from IL-6-targeted therapeutic intervention, which is expected to inhibit both Th17 cell differentiation and cytotoxic CD8 T cell activation in the pathological mechanisms of irAE.

The first and foremost possible limitation of the current study is its retrospective nature in human sample collection from a limited number of institutes. Therefore, potential biases, such as selection bias and reporting bias, cannot be excluded, and functional analysis of CD8 T cells in skin and blood has yet to be completed. In addition, there are potentially some differences in the pathogenic mechanisms of psoriasis-like dermatitis between PD-1-deficient mice and anti-PD-1 antibody-treated mice/

humans. These include our results that serum levels of IL-23 were significantly elevated in anti-PD-1 antibody-treated mice with psoriasis-like dermatitis compared to control mice, while the levels were equal between PD-1-deficient mice and WT mice. Further, our study did not directly addressed if PD-1 signal blockade on CD4 T cells could have some effects, as was reported in a murine model of virus infection where the effects on CD4 T cells alter CD8 T-cell function through PD-1 signal blockade[45]. Moreover, the exact source of IL-6 is yet to be determined, even though IL-6 mRNA levels in CD45-positive epidermal cells, a potential cell population identified in the current study, were significantly elevated in PD-1-deficient mice compared to WT mice. Further prospective studies are needed to clarify those findings. Despite the limitations, data from the current study highlighted the unique characteristics of PD-1 signal blockade-induced psoriasis-like dermatitis, most strikingly the significance of strong correlation between the enhanced IL-6 production and the dermatitis development, indicating the potential significance of IL-6-targeting for therapeutic intervention.

In summary, IL-6 plays important roles during disease development of PD-1 signal blockade-induced psoriasis-like dermatitis. Moreover, PD-1 expressed on CD8 T cells is responsible for the regulation of skin inflammation. Blockade of IL-6 signaling decreases inflammation in PD-1 signal inhibition-provoked psoriasis-like dermatitis, and specifically, causes a reduction in the levels of Th17-related cytokines in a murine model of IMQ-induced psoriasis-like dermatitis. Thus, these findings highlight the potential significance of IL-6-targeting for therapeutic intervention of PD-1 signal blockade-induced psoriasis-like dermatitis in humans.

## Methods

**Human sample collection**. Formalin-fixed paraffin-embedded (FFPE) skin biopsy samples were obtained from melanoma ($n = 3$), renal cell carcinoma ($n = 2$), gastric cancer ($n = 1$) and lung cancer ($n = 1$) patients with anti-PD-1 antibody-induced psoriasis-like dermatitis ($n = 7$, totally), and idiopathic psoriasis patients ($n = 6$), who visited Tsukuba University Hospital (Japan) and Mito Saiseikai General Hospital (Japan) from 2014 to 2018. Serum samples were collected post-treatment from melanoma ($n = 25$), renal cell carcinoma ($n = 1$), and lung cancer ($n = 1$) patients treated with anti-PD-1 antibody at Tsukuba University Hospital (Japan) from 2014 to 2019 ($n = 27$), including eight patients who developed psoriasis-like dermatitis after the treatment. Patient clinical data were retrospectively reviewed from their medical records.

**Mice**. Wild-type (WT) C57BL/6J male mice originally from the Jackson Laboratories were purchased from Charles River Japan. PD-1-knockout (PD-1$^{-/-}$) mice were provided by Dr. Tasuku Honjo (Kyoto University, Japan). We generated mice with a PD-1 allele mutated by the insertion of two loxP sites flanking parts of the promoter region (PD-1$^{fl/fl}$ mice) using a CRISPR-Cas9 system at Laboratory Animal Resource Center, University of Tsukuba. PD-1$^{fl/fl}$ mice develop normally indicating that the insertion of the loxP sites does not significantly interfere with regulation of the *PD-1* gene. Floxed heterozygous PD-1$^{fl/+}$ and heterozygous CD8$^{cre}$ mice (C57BL/6-Tg (Cd8a-cre)1ltan/J, Jackson Laboratories) were crossed to generate double heterozygous PD-1$^{fl/+}$; CD8$^{cre}$ mice, which were bred with homozygous PD-1$^{fl/fl}$ mice to produce conditional PD-1 homozygous (PD-1$^{fl/fl}$CD8$^{cre}$), PD-1 conditional knockout (PD-1-cKO), mice, and their PD-1 heterozygous (PD-1$^{fl/+}$CD8$^{cre}$) littermates (Littermate Ctrl, Fig. 3). The primer sequences used for genotyping of CD8$^{cre}$, PD-1$^{-/-}$, and PD-1$^{fl}$ are listed in Supplemental Table 1. We confirmed the complete deletion of PD-1 expression specifically in the CD8 T cell population (CD3$^+$CD8$^+$ lymphocytes) in lymph nodes of PD-1-cKO mice by flow cytometry (Supplemental Fig. 1). C57BL/6 background male mice, 8 to 12-weeks-old, were maintained in specific pathogen-free conditions and used for all experiments.

**Murine model of psoriasis-like dermatitis**. In order to replicate a modified murine model of IMQ-induced psoriasis-like dermatitis[30], 3.5% IMQ cream diluted from 5% IMQ cream (Beselna®; Mochida Pharmaceuticals) with vehicle control cream (Vanicream®; Pharmaceutical Specialties) (62.5 mg IMQ total, which was a lower dose compared to the dose used in a conventional model of IMQ-induced psoriasis-like dermatitis) was applied topically on a daily basis to the shaved back and both ears for 5 or 7 consecutive days. Control mice were treated with the vehicle control cream only.

**Scoring system for evaluating the severity of skin inflammation**. To score the severity of inflammation of the back skin, an objective scoring system mimicking the Psoriasis Area and Severity Index (PASI) score for psoriasis patients was used as in a previous study[46], in which independent scores of erythema, scaling, and thickening with a scale from 0 to 4 (0, none; 1, slight; 2, moderate; 3, marked; 4, very marked) were cumulated (ranged from 0 to 12). The ear thickness was measured using a micrometer (Mitutoyo).

**IL-6 blockade**. An anti-interleukin-6 receptor (anti-IL-6R) blocking antibody (MR16-1, Chugai Pharmaceuticals), which is a rat IgG1 monoclonal antibody against murine IL-6Rα chain, was injected intravenously at the dose of 2 mg per mouse prior to IMQ application at day 0. IgG isotype control (MP Biomedicals™) was used as a control.

**Histological analysis**. All FFPE human skin biopsy samples and murine ear skin samples were sectioned into 2 and 4-μm-thick slides, respectively, and subsequently undergone hematoxylin-eosin (HE) staining. Human FFPE skin samples were stained immunohistologically with anti-human CD8 and anti-human CD4 monoclonal antibodies (clone C8/144B and 4B12, respectively, Nichirei Biosciences) using an automatic slide stainer according to the manufacturer's instructions. The numbers of epidermal-infiltrating cells per sample (magnification, ×400) were counted. Murine ear FFPE samples were stained immunohistochemically with primary anti-CD3 (clone SP7, diluted 1:100, Abcam) and anti-CD8a (clone 4SM15, diluted 1:400, eBioscience) monoclonal antibodies, fluorescent-labeled secondary antibodies (Alexa Fluor® 488-labeled goat anti-rabbit IgG, and Alexa Fluor® 555-labeled goat anti-rat IgG, Abcam), and 4′,6-diamidino-2-phenylindole (DAPI) to detect the nucleus, by standard immunohistochemical staining techniques. A fluorescence microscope (BZ-X700, Keyence) was used for observation and to count the number of infiltrating cells per sample (magnification, ×400).

**Blood sample assay system**. Murine blood samples were collected using the submandibular bleeding method and serum samples were subsequently isolated. Human and murine serum samples were immediately stored at $\leq -20\,°C$ for later use. In order to analyze cytokine (human and murine IL-6, murine IL-23A, and IL-17A) serum levels, the MILLIPLEX® MAP Kit (Merck Millipore) using Bio-Plex® Luminex 200 multiplex assay system (Bio-Rad) was employed according to the manufacturer's protocol. Human serum levels of sIL-6 and sIL-6Rα were measured using Enzyme-Linked Immuno Sorbent Assay (ELISA) kit (Duoset and Quantikine; R&D systems) according to the manufacturer's protocol.

**Quantitative reverse transcription-polymerase chain reaction (qRT-PCR)**. Total RNA was extracted from the murine ear samples using Trizol Reagent (Invitrogen). RNA concentrations were quantified and the $OD_{260/230}$ and the $OD_{260/280}$ ratio of the RNA samples were confirmed to be more than 1.8 and 1.6, respectively, with the NanoDrop ND-1000 (peqLab Biotechnologie GmbH). Complementary DNA (cDNA) was synthesized with a High-Capacity cDNA Reverse Transcription Kit (Thermo Fisher) according to the manufacturer's instructions. Messenger RNA (mRNA) expression levels were detected by PCR amplification of cDNA using the QuantStudio™ 5 Real-Time PCR Systems (Applied Biosystems) with PrimeTime® Gene Expression Master Mix and Prime Tim qPCR predesigned primers (Integrated DNA Technologies) listed in Supplemental Table 1. All qRT-PCR analyses were performed in triplicate. Amplification products were quantified by the comparative CT method. The mRNA level of each gene was normalized to that of *glyceraldehyde-3-phosphate dehydrogenase* (*GAPDH*).

**Skin separation**. Murine ear skin samples were treated with 0.25% trypsin (FUJIFILM Wako Pure Chemical Corporation) solution for 40 minutes at 37 °C in order to separate the epidermis and dermis. After washing two times with phosphate-buffered saline without $Ca^{2+}$ and $Mg^{2+}$ and passing through a 70 μm cell strainer, dissociated epidermal cells were then separated into CD45$^-$ single cells (keratinocytes) and CD45$^+$ single cells using MACS® cell separation technology with CD45 MicroBeads beads (Miltenyi Biotec) according to the manufacturer's instructions. The positive selected fractions and the negative sorted fractions contained more than 95% and less than 1% of CD45-positive cells, respectively, by flow cytometry (data not shown).

**Flow cytometry**. Draining lymph nodes (dLNs) were harvested and single-cell suspensions were prepared. For the exclusion of dead cells, the Zombie fixable viability kit (BioLegend) was used. Cells were incubated in FACS staining buffer (PBS containing 1% BSA and 5 mM EDTA) with anti-FcγIII/II receptor antibody (BD), and anti-CD45 (30-F11, BioLegend), anti-CD4 (Gk1.5, BioLegend), anti-CD8 (53–6.7, BioLegend), anti-CD3e (145-2C11, BioLegend), anti-B220 (RA3-6B2, eBioscience), and anti-PD-1 (29F.1A12, BioLegend) antibodies. For intracellular IFN-γ and Gzm B staining, cells were stimulated with 25 ng/ml PMA and 1 μg/ml Ionomycin in RPMI 1640 medium supplemented with 10% fetal bovine serum, 2 mM L-glutamine, 100 U/ml penicillin, and 100 μg/ml streptomycin (complete RPMI) with monensin (Golgi Stop, BD). After five hours of incubation, cell surface staining was followed by

intracellular cytokine staining using the Fix/Perm Kit (BD) in accordance with the manufacturer's instructions with anti–IFN-γ (XMG1.2, BD) and anti-Gzm B (NGZB, eBioscience) antibodies. Fluorescence-minus-one controls were used as negative controls. Cells were acquired on the Gallios (Beckman-Coulter) and data were analyzed using the FlowJo software (v7.6.5).

**Statistics and reproducibility.** The differences between the groups were evaluated by Student's *t* test, Mann–Whiney *U* test or two-way ANOVA using GraphPad Prism 7.0 Software. A value of $P < 0.05$ was considered to be statistically significant. We repeated at least twice experiments and the exact sample size (n) for each experiment appear in the figure legend.

**Study approval**. All patients provided written, informed consent in compliance with the approval by the Institutional Ethics Committee at the University of Tsukuba Hospital (number: H28-045 and H30-256). All animal experiments were approved by the Animal Experiment Committee of the University of Tsukuba (Permit Number: 17–137), and performed in accordance with the Guide for the Care and Use of Laboratory Animals of the University of Tsukuba.

**Reporting summary**. Further information on research design is available in the Nature Research Reporting Summary linked to this article.

## Data availability
Raw data for graphs can be found in Supplementary Data 1. All other data are available within the manuscript files or from the corresponding author upon reasonable request.

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

## Acknowledgements
The authors would like to thank Miwako Shobo, Hideko Sakurai, and Yuriko Hirota for excellent technical support, and Immunology Science Editors for English language editing (www.immunologyscienceeditors.com).

## Author contributions
R.T. and N.O. designed research studies. Y.I., N.K., A.K., and R.T. conducted experiments and acquired data. R.T., N.O., Y.N., R.W., Y.F., and M.F. analyzed data. M.K. provided human samples, S.M. and S.T. provided PD-1 floxed mice, respectively, and critically edited the manuscript. R.T. and N.O. wrote the manuscript. All authors approved the manuscript.

## Competing interests
The authors declare no competing interests.
