## [Peer Review File · Communications Biology]

Reviewers' comments:

Reviewer #1 (Remarks to the Author):

In the manuscript "IL-6 targeted therapy for psoriasis exacerbated by inhibition of PD-1 signaling on CD8 T cells", the authors propose that immune related adverse event, particularly psoriasis-like dermatitis in melanoma patients undergoing PD-1 blockade might be due to increased IL6 levels. They use observations of elevated IL6 and CD8 Tcells in idiopathic psoriasis versus melanoma patients (aPD1 blockade treated) developing psoriasis-like dermatitis to substantiate their hypothesis via mouse models.

The manuscript is well written. There are labeling errors/missing info on axis of graphs with mRNA qPCR results in all figures that need to be addressed.

Major comments:

1. Genetic deficiency of PD1 in mice might not best reflect immunotherapy-associated biology. It would be important to test in B16 melanoma mouse model, does PD1 blockade leads to psoriasis-like dermatitis. In such a setting, would IL6 drive the adverse event? If so would anti IL6R blockade improve the pathology?
2. Important to identify the source of IL6. CD8 Tcells likely are not source of IL6. Can authors show which innate immune cell in skin is likely source of IL6 or is perturbed in IMQ model in WT v PD1 KO?
3. In Fig 1c: it would be worth to compare CD8/CD4 ratio in skin and in blood of patients undergoing anti PD1 therapy (those who develop irAE versus those who do not). Just like fig 1D
4. In humans, as per authors (ref25), only IL6 was changed and not IL17, IFN γ or IL23. So it seems like, the PD1 genetic knockout or cKO that show effect on other cytokines including IL6, might be slightly different from patients. Anti PD1 blockade expt in mice would be important to distinguish developmental roles of PD1 genetic deficiency versus acute antibody mediated checkpoint blockade.
5. In mouse models CD8 cre would potentially delete PD1 in double positive thymocytes so in essence even CD4 T cells might lack PD1. Can authors use CD8 T cell depletion or show that the phenotype is not affected by CD4 Tcells lacking PD1?
6. Would IL17 blockade that is approved in humans with psoriasis work in the context of PD1 KO mice and IMQ-induced psoriasis? Can authors compare IL6R blockade with anti IL17A/F neutralization?
7. For fig4-6, can authors show elevated IL6 levels in serum for mice that lack PD1, either KO or cKO versus control littermate after IMQ treatment and not at baseline.

Reviewer #2 (Remarks to the Author):

In this manuscript, authors elucidated the underlining mechanism of PD-1 signal blockade-induced psoriasis-like dermatitis and suggested that it is an IL-6 dependent pathology. Authors also suggested PD-1 signal blockade-induced psoriasis-like dermatitis can potentially benefit from IL-6-targeted therapeutic intervention.

Authors used different mouse strains and a blocking antibody to prove the hypothesis. The result is well presented; however, cross-check is required to make sure all figure legends give the correct description of the data

Major comment

1. Overexpression of PD-1 in IL-17A producing T cells was previously reported in IMQ-treated mice, which helped IL-17 production (Kim et.al JACI 2015 <https://doi.org/10.1016/j.jaci.2015.11.021>). However, in this study, mice that lack PD-1 has high IL-17A mRNA levels (imply high IL-17 production) under IMQ-treatment. Can the authors comment on the discrepancy?

2. The effectiveness of anti-IL6 therapies in psoriasis remains queried. Treatment with tocilizumab, approved for use in RA, has been reported in different studies to induce psoriasis in patients with RA with no history of psoriasis. Fritz et.al. 2017 also suggested that therapeutic blockade of IL-6 function in humans may lead to excess production of other proinflammatory cytokines in skin. Such overcompensation would likely abrogate the therapeutic benefits of anti-IL-6 agents in psoriasis. The authors focused on a specific PD-1 signal blockade-induced psoriasis-like dermatitis, which demonstrated a feature of high CD8 T cells from both patients sample and IMQ-dependent psoriasis model. The underlying pathologies of the PD-1 signal blockade-induced psoriasis-like dermatitis and psoriasis in human are both IL-17 driven, and high IL-6 level and CD8 T cells are observed. Why that anti-IL-6 demonstrated efficacy in the PD-1-dependent manner but not in normal psoriasis

3. A robust increase in CD8 T cells is well documented in the IMQ-induced psoriasis model. However, there was no obvious increase on the WT mice and control littermate compared to the PD1^{-/-} and PD1-cKO mice in this study. (Fig3B and Figure 4J)

4. Can the authors detect a high level of soluble IL-6 receptor in serum samples collected from anti-PD-1 antibody-treated cancer patients who developed psoriasis-like dermatitis? Alternatively, can IL-6^Rhigh CD8⁺ Cells be detected in these patient's blood (a subset of effector T cells present in asthma or viral infection)?

Minor comment

Line 49 – 51: Statement is contradicting. “most of which are mild (low-grade) and manageable with topical steroids. However, two-thirds of patients reportedly required systemic corticosteroids for the treatment of eruptions.”

Line 52: “19% of patients discontinued cancer-immunotherapy due to irAEs”. Please specify if the irAEs mentioned are cutaneous.

Line 147-148: Was high CD8a and IFN γ mRNA detected in the serum of PD-1-cKO mice? Figure legend of 4H indicated the mRNA was measured from ear skin samples.

Fig1B: Reviewer agrees that there was an increase in CD8 cells in anti-PD-1 antibody-induced psoriasis-like dermatitis compared to psoriasis from the representative images. However, these images also indicated more CD4 than CD8 cells in both cases. Why the median of the CD8/CD4 ratio is over 1?

Fig2: Figure Legend needs to be revised. There are figure 2I-J in the figure legend which cannot be found in the figure.

Fig3. The number of CD8a positive cell per area/length should be used in Fig3C but not the total number.

Fig4: Figure legend needs to be revised. Please pay attention to the consistency of the numbering (A-

J).

qRT-PCR result: Y-Axis title was missing for all result.

Point-by-point responses to the reviewer comments

Reviewer 1

Comment #1. Genetic deficiency of PD1 in mice might not best reflect immunotherapy-associated biology. It would be important to test in B16 melanoma mouse model, does PD1 blockade leads to psoriasis-like dermatitis. In such a setting, would IL6 drive the adverse event? If so would anti IL6R blockade improve the pathology?

Response #1: We thank the reviewer for bringing up this point. We conducted an experiment using a B16 melanoma mouse model, in which B16F10 melanoma cells (B16 melanoma) were inoculated into the backs of in both WT and PD-1^{-/-} mice, to investigate whether the presence of a malignant tumor results in PD-1 blockade-induced psoriasis-like dermatitis. First, this model did not induce psoriasis-like dermatitis spontaneously in WT or PD-1^{-/-} mice. Moreover, the presence of B16 melanoma did not lead to a significant difference in PASI score or ear swelling neither in WT or PD-1^{-/-} mice. These results indicate that PD-1 signal blockade does not collaborate with the presence of B16 melanoma in the murine model of imiquimod-induced psoriasis-like dermatitis (3rd paragraph of page 8, and Supplemental Figure 3).

Comment #2. Important to identify the source of IL6. CD8 T cells likely are not source of IL6. Can authors show which innate immune cell in skin is likely source of IL6 or is perturbed in IMQ model in WT v PD1 KO?

Response #2: We fully agree with the reviewer and are grateful for the useful suggestions, which we have done our best to implement. As various hematopoietic and non-hematopoietic cells are known to produce IL-6, it is difficult to narrow down the IL-6-producing cell type. Therefore, we first investigated which layer of the skin exhibits differential IL-6 production between WT and PD-1^{-/-} mice. Separation of skin into keratinocytes, CD45-positive epidermal cells, and dermal cells revealed significantly increased IL-6 mRNA expression in CD45-positive epidermal cells of PD-1^{-/-} mice compared to WT mice. Moreover, further analysis of the levels of CD45-positive epidermal cells revealed an increased total number in PD-1^{-/-} mice. In addition, neutrophils specifically infiltrated into the epidermis of PD-1^{-/-} mice. Thus, increased epidermal CD45⁺ cells with neutrophil infiltration appear to contribute to the difference

in IL-6 production between PD-1^{-/-} and WT mice (1st and 2nd paragraphs of page 11, and Supplemental Figure 4).

Comment #3. In Fig 1c: it would be worth to compare CD8/CD4 ratio in skin and in blood of patients undergoing anti PD1 therapy (those who develop irAE versus those who do not). Just like fig 1D

Response #3: We fully agree with this suggestion. Regretfully, due to the retrospective nature of this study we were not able to collect hematocyte samples. Meanwhile, the results from a prospective observational study investigating the efficacy of anti-PD-1 antibody (nivolumab) in a real-world setting will be completed soon. In this study selective biomarkers are investigated (Clinical study of the efficacy, evaluation, and changes in biomarkers in response to nivolumab treatment of an advance malignant melanoma; CREATIVE, UMIN000016608). We expect the results from this study will address this issue. In response to the reviewer's comment, we have described this point as a limitation of the current manuscript (2nd paragraph of page 20).

Comment #4. In humans, as per authors (ref25), only IL6 was changed and not IL17, IFN γ or IL23. So it seems like, the PD1 genetic knockout or cKO that show effect on other cytokines including IL6, might be slightly different from patients. Anti PD1 blockade expt in mice would be important to distinguish developmental roles of PD1 genetic deficiency versus acute antibody mediated checkpoint blockade.

Response #4: We thank the reviewer for pointing out this important aspect. Because the effects of PD-1 blockade, either by genetic knockout or mAb treatment, have been previously shown to promote psoriasiform skin inflammation induced by topical IMQ treatment (Imai *et al. J Immunol*, 2015), we only included our results from PD-1^{-/-} mice. To respond to the suggestion by the reviewer, we performed additional experiments using an anti-PD-1 blocking monoclonal antibody (mAb) and have included the results in Supplemental Figure 2.

First, we confirmed that wild-type (WT) mice treated with anti-PD-1 mAb exhibit severe imiquimod-induced psoriasis-like dermatitis clinically, as assessed by ear swelling and PASI score, when compared to mice treated with isotype control mAb. In addition, histological analysis also showed enhanced epidermal thickness and increased

numbers of neutrophilic micro-abscesses in anti-PD-1 mAb-treated mice than in isotype control mAb-treated animals (2nd paragraph of page 8, and Supplemental Figure 2A-E).

Further, we investigated the number of CD8 T cells infiltrating into the epidermis by immunohistochemical analysis of murine ear skin samples, and demonstrated an increased number in the anti-PD-1 mAb-treated group compared to the control mAb-treated group (2nd paragraph of page 9, and Supplemental Figure 2F).

Finally, analysis of psoriasis-related cytokine production, including IL-6, revealed the same trend in the anti-PD-1 mAb treated mice as in PD-1^{-/-} mice (1st paragraph of page 11, and Supplemental Figure 2G).

Collectively, these data suggest that blockade of PD-1 signaling, either by genetic knockout or blocking antibody treatment, promotes psoriasis-like dermatitis.

Comment #5. In mouse models CD8^{cre} would potentially delete PD1 in double positive thymocytes so in essence even CD4 T cells might lack PD1. Can authors use CD8 T cell depletion or show that the phenotype is not affected by CD4 T cells lacking PD1?

Response #5: We think that this is an impactful question, especially in the field of basic immunology research. We applied IMQ to the skin and harvested draining lymph nodes from CD8^{cre}PD-1^{flox/flox} (PD-1 cKO) and WT littermates. Flow cytometric analysis clearly showed complete deletion of PD-1 expression specifically in the CD8 T cell population, but not in the CD4 T cell nor B cell populations, in the draining lymph nodes of PD-1-cKO mice (Figure 4B).

Comment #6. Would IL17 blockade that is approved in humans with psoriasis work in the context of PD1 KO mice and IMQ-induced psoriasis? Can authors compare IL6R blockade with anti IL17A/F neutralization?

Response #6: The reviewer makes an excellent point. A previous study showed that IL-17A neutralization could be a treatment option for IMQ-induced psoriasis-like dermatitis under PD-1 signaling blockade. Here, we treated IMQ-induced psoriasis-like dermatitis in PD-1^{-/-} mice with anti-IL-17A blocking antibody and compared the results with anti-IL-6R antibody. On day 7, anti-IL-17A and anti-IL-6R antibody-treated PD-1^{-/-} mice showed clinical and histologic improvement of psoriasis-like dermatitis. However, treatment with anti-IL-6R antibody resulted in more rapid improvement of dermatitis (on day 3) compared with anti-IL17A antibody treatment. Hence, targeting

IL-6 may provide rapid and marked improvement in PD-1-blockade induced psoriasis-like dermatitis (2nd paragraph of page 13 to 2nd paragraph of page14 , and Supplemental Figure 5).

Comment #7. For fig4-6, can authors show elevated IL6 levels in serum for mice that lack PD1, either KO or cKO versus control littermate after IMQ treatment and not at baseline.

Response #7: We appreciate this question and have added the baseline data from both WT and PD-1^{-/-} mice in our revision. There was not any significant difference in baseline serum levels of cytokines between the two groups (revised Figure 5H).

Reviewer 2

Comment #1. Overexpression of PD-1 in IL-17A producing T cells was previously reported in IMQ-treated mice, which helped IL-17 production (Kim *et.al* JACI 2015 <https://doi.org/10.1016/j.jaci.2015.11.021>). However, in this study, mice that lack PD-1 has high IL-17A mRNA levels (imply high IL-17 production) under IMQ-treatment. Can the authors comment on the discrepancy?

Response #1: We thank the reviewer for bringing up this point. PD-1 is expressed on stimulated T cells, in other words, PD-1 is an activation marker. The important point of the results provided by Kim *et al.* is the remarkable production of IL-17A by PD-1^{hi} $\gamma\delta$ T cells, which could be inhibited by the interaction between PD-1 and its ligand in IMQ-induced psoriasis-like dermatitis. Moreover, another report (Imai *et al.* J Immunol, 2015) demonstrated the promotion of IL-17A production by PD-1^{-/-} or anti-PD-1 blocking antibody-treated $\gamma\delta$ T cells in psoriasis-like dermatitis. Taken together, these reports confirm our finding that the PD-1 signaling pathway has an inhibitory effect on T cells.

Comment #2. The effectiveness of anti-IL6 therapies in psoriasis remains queried. Treatment with tocilizumab, approved for use in RA, has been reported in different studies to induce psoriasis in patients with RA with no history of psoriasis. Fritz *et.al.* 2017 also suggested that therapeutic blockade of IL-6 function in humans may lead to excess production of other proinflammatory cytokines in

skin. Such overcompensation would likely abrogate the therapeutic benefits of anti-IL-6 agents in psoriasis. The authors focused on a specific PD-1 signal blockade-induced psoriasis-like dermatitis, which demonstrated a feature of high CD8 T cells from both patients sample and IMQ-dependent psoriasis model. The underlying pathologies of the PD-1 signal blockade-induced psoriasis-like dermatitis and psoriasis in human are both IL-17 driven, and high IL-6 level and CD8 T cells are observed. Why that anti-IL-6 demonstrated efficacy in the PD-1-dependent manner but not in normal psoriasis.

Response #2: We thank the reviewer for pointing out this important aspect. It is well-known that conventional psoriasis is a chronic inflammatory disease depending on Th17 production of IL-17A, IL-17F and IL-22, which makes these the main target cells for treatment. Blockade of IL-6 might induced paradoxical reactions in the long-term, as the reviewer pointed out. On the other hand, PD-1 signal blockade-induced irAEs, including psoriasis-like dermatitis, are usually acute/subacute diseases during cancer treatment. Thus, the aim of irAEs treatment is to resolve or reduce the severity of irAE symptoms rapidly to enable shorter interruptions of cancer treatments. IL-6 is the main cytokine involved in the induction of acute phase inflammation, which seems to play an important role in development of PD-1 signal blockade-induced psoriasis-like dermatitis, as shown in previous clinical reports and the present study. Moreover, data from the murine model indicates that IL-6-targeted therapeutic intervention might inhibit both Th17 cell differentiation and cytotoxic CD8 T cell activation. Since CD8 T cell activation and expansion in the epidermis are unique characteristics of PD-1 signal blockade-induced psoriasis-like dermatitis, this dual inhibition appears to be important. Moreover, a previous report showed that one dose of anti-IL-6R antibody (tocilizumab; TCZ) resolved the symptoms of steroid-refractory irAE in half of patients (Stroud CR *et al. J Oncol Pharm Pract*, 2019). According to a literature summary of TCZ-induced paradoxical psoriasis-like eruption, the median time from TCZ initiation to eruption development is 9 weeks (Hayakawa *et al. Rheumatol int*, 2019). Therefore, the paradoxical induction of psoriasis-like eruptions by IL-6 blockade likely occurs only after long-term use and thus, short-term use should be relatively safe. A modification of the TCZ dose could improve the development of this psoriasis-like dermatitis, although the underlying mechanisms are unknown (Hayakawa *et al. Rheumatol int*, 2019).

In summary, short-term use of TCZ may be reasonable and provide improved quality of life for irAE patients. Future clinical studies of TCZ treatment in patients with

PD-1 signal blockade-induced psoriasis-like dermatitis will be necessary to definitively resolve this issue.

We have added this information to the Discussion section of our revised manuscript (2nd paragraph of page 19).

Comment #3. A robust increase in CD8 T cells is well documented in the IMQ-induced psoriasis model. However, there was no obvious increase on the WT mice and control littermate compared to the PD1-/- and PD1-cKO mice in this study. (Fig3B and Figure 4J)

Response #3: We thank the reviewer for raising this issue. The most likely reason is that we used a low-dose of IMQ (3.5% cream) compared to that in the conventional model of IMQ-induced psoriasis-like dermatitis. It was previously demonstrated (Imai *et al. J Immunol*, 2015) that the difference in severity of IMQ-induced psoriasis-like dermatitis between WT mice and PD-1^{-/-} mice was apparent with a low-dose of IMQ (3.5% cream), but not a conventional dose of IMQ (5% cream), in B6-background mice. Because the PD-1/PDL pathway regulates CD8 T cell activation, it is possible that CD8 T cell infiltration in WT mice treated with a low-dose of IMQ may be inhibited by this pathway, as the inflammation was abrogated by PD-1 blockade in the present study.

We have described the modified method for IMQ-induced psoriasis-like dermatitis in the Methods section of our revised manuscript (2nd paragraph of page 22).

Comment #4. Can the authors detect a high level of soluble IL-6 receptor in serum samples collected from anti-PD-1 antibody-treated cancer patients who developed psoriasis-like dermatitis? Alternatively, can IL-6Ralphahigh CD8+ Cells be detected in these patient's blood (a subset of effector T cells present in asthma or viral infection)?

Response #4: We thank the reviewer for this helpful recommendation. We detected soluble IL-6 receptor alpha (sIL-6R α) in serum samples from anti-PD-1 antibody-treated cancer patients. The level of sIL-6R α in psoriasis-like dermatitis patients was not significantly different from that of patients without irAE. However, regression analysis revealed a strong correlation between sIL-6 and sIL-6R α serum levels (R-squared correlation of 0.4484) in patients with psoriasis-like dermatitis, which was not observed in patients without irAE (R-squared correlation < 0.0001, Supplemental Figure 1). In previous reports the serum levels of sIL-6R α were scattered, which may be due more to

the presence of cancer than to the sIL-6 serum level (1st paragraph of page 7, and 2nd paragraph of page 18).

Minor comment

We thank the reviewer for these pertinent comments. A brief answer to each comment is listed below.

Line 49 – 51: Statement is contradicting. “most of which are mild (low-grade) and manageable with topical steroids. However, two-thirds of patients reportedly required systemic corticosteroids for the treatment of eruptions.”

Line 52: “19% of patients discontinued cancer-immunotherapy due to irAEs”. Please specify if the irAEs mentioned are cutaneous.

The statement has been revised (1st paragraph of page 4).

Line 147-148: Was high CD8a and IFN γ mRNA detected in the serum of PD-1-cKO mice? Figure legend of 4H indicated the mRNA was measured from ear skin samples.

We use ear skin samples. The figure has been revised (Figure 4H).

Fig1B: Reviewer agrees that there was an increase in CD8 cells in anti-PD-1 antibody-induced psoriasis-like dermatitis compared to psoriasis from the representative images. However, these images also indicated more CD4 than CD8 cells in both cases. Why the median of the CD8/CD4 ratio is over 1?.

Thank you for pointing this out. Along with the revision, we found that the images were labeled in reverse, and thus we have corrected the label. In addition, we outlined the border between the epidermis and dermis with a yellow dotted line (Figure 1B). In both psoriasis-like dermatitis and psoriasis, most CD4 cells were present in dermis, including the prolonged papillae of dermis, while CD8 cells were present in both the dermis and epidermis. Hence, the median CD8/CD4 ratio of intraepidermal lymphocytes was over 1 (CD8 > CD4).

Fig2: Figure Legend needs to be revised. There are figure 2I-J in the figure legend which cannot be found in the figure.

The figure legend has been revised.

Fig3. The number of CD8a positive cell per area/length should be used in Fig3C but not the total number.

The figure has been revised to the number of CD8a positive cells per high power field (HPF).

REVIEWERS' COMMENTS:

Reviewer #1 (Remarks to the Author):

The authors have addressed my major concerns diligently. Most of the experiments done do justify their major conclusions and take home message. It is important they acknowledge some of their study limitations and caveats in a small discussion section if possible, which would show to the audience their insights and appreciation of future work that needs to be done.

Reviewer #2 (Remarks to the Author):

The Authors have addressed all reviewer's concerns.

In Suppl 1, reviewer acknowledges that there was no significant difference on the sIL-6Ra level between No irAE and anti-PD1 groups in Fig S1A. The correlation between sIL-6 and sIL-6Ra however cannot provide further supportive information and could be excluded.

In page 5, line 8, there is a calculation error. 19/103 should be 18%

Point-by-point responses to the reviewer comments

Reviewer #1:

> The authors have addressed my major concerns diligently. Most of the experiments done do justify their major conclusions and take home message. It is important they acknowledge some of their study limitations and caveats in a small discussion section if possible, which would show to the audience their insights and appreciation of future work that needs to be done.

We thank the reviewer for this favorable comment and additional suggestion. We have added several sentences stating the limitations of our study to the Discussion (page 20-21).

Reviewer #2:

> The Authors have addressed all reviewer's concerns.

We thank the reviewer for this favorable comment.

> In Suppl 1, reviewer acknowledges that there was no significant difference on the sIL-6Ra level between No irAE and anti-PD1 groups in Fig S1A. The correlation between sIL-6 and sIL-6Ra however cannot provide further supportive information and could be excluded.

We thank the reviewer for this helpful recommendation. In response to this we have stated that there is no significant difference in the serum levels of sIL-6R α between the two groups, and deleted our analysis using the R-squared correlation.

> In page 5, line 8, there is a calculation error. 19/103 should be 18%

We have corrected this error.